# Near-optimal Reinforcement Learning in Factored MDPs

**Ian Osband**
Stanford University
iosband@stanford.edu

**Benjamin Van Roy**
Stanford University
bvr@stanford.edu

## Abstract

Any reinforcement learning algorithm that applies to all Markov decision processes (MDPs) will suffer $\Omega(\sqrt{SAT})$ regret on some MDP, where $T$ is the elapsed time and $S$ and $A$ are the cardinalities of the state and action spaces. This implies $T = \Omega(SA)$ time to guarantee a near-optimal policy. In many settings of practical interest, due to the curse of dimensionality, $S$ and $A$ can be so enormous that this learning time is unacceptable. We establish that, if the system is known to be a *factored* MDP, it is possible to achieve regret that scales polynomially in the number of *parameters* encoding the factored MDP, which may be exponentially smaller than $S$ or $A$. We provide two algorithms that satisfy near-optimal regret bounds in this context: posterior sampling reinforcement learning (PSRL) and an upper confidence bound algorithm (UCRL-Factored).

## 1 Introduction

We consider a reinforcement learning agent that takes sequential actions within an uncertain environment with an aim to maximize cumulative reward [1]. We model the environment as a Markov decision process (MDP) whose dynamics are not fully known to the agent. The agent can learn to improve future performance by exploring poorly-understood states and actions, but might improve its short-term rewards through a policy which exploits its existing knowledge. Efficient reinforcement learning balances exploration with exploitation to earn high cumulative reward.

The vast majority of efficient reinforcement learning has focused upon the *tabula rasa* setting, where little prior knowledge is available about the environment beyond its state and action spaces. In this setting several algorithms have been designed to attain sample complexity polynomial in the number of states $S$ and actions $A$ [2, 3]. Stronger bounds on regret, the difference between an agent's cumulative reward and that of the optimal controller, have also been developed. The strongest results of this kind establish $\tilde{O}(S\sqrt{AT})$ regret for particular algorithms [4, 5, 6] which is close to the lower bound $\Omega(\sqrt{SAT})$ [4]. However, in many setting of interest, due to the curse of dimensionality, $S$ and $A$ can be so enormous that even this level of regret is unacceptable.

In many practical problems the agent *will* have some prior understanding of the environment beyond *tabula rasa*. For example, in a large production line with $m$ machines in sequence each with $K$ possible states, we may know that over a single time-step each machine can only be influenced by its direct neighbors. Such simple observations can reduce the dimensionality of the learning problem exponentially, but cannot easily be exploited by a *tabula rasa* algorithm. Factored MDPs (FMDPs) [7], whose transitions can be represented by a dynamic Bayesian network (DBN) [8], are one effective way to represent these structured MDPs compactly.

Several algorithms have been developed that exploit the known DBN structure to achieve sample complexity polynomial in the *parameters* of the FMDP, which may be exponentially smaller than $S$ or $A$ [9, 10, 11]. However, these polynomial bounds include several high order terms. We present two algorithms, UCRL-Factored and PSRL, with the first near-optimal regret bounds for factored MDPs. UCRL-Factored is an optimistic algorithm that modifies the confidence sets of UCRL2 [4] to take advantage of the network structure. PSRL is motivated by the old heuristic of Thompson sampling [12] and has been previously shown to be efficient in non-factored MDPs [13, 6]. These algorithms are descibed fully in Section 6.

Both algorithms make use of approximate FMDP planner in internal steps. However, even where an FMDP can be represented concisely, solving for the optimal policy may take exponentially long in the most general case [14]. Our focus in this paper is upon the statistical aspect of the learning problem and like earlier discussions we do not specify which computational methods are used [10]. Our results serve as a reduction of the reinforcement learning problem to finding an approximate solution for a given FMDP. In many cases of interest, effective approximate planning methods for FMDPs do exist. Investigating and extending these methods are an ongoing subject of research [15, 16, 17, 18].

## 2   Problem formulation

We consider the problem of learning to optimize a random finite horizon MDP $M = (\mathcal{S}, \mathcal{A}, R^M, P^M, \tau, \rho)$ in repeated finite episodes of interaction. $\mathcal{S}$ is the state space, $\mathcal{A}$ is the action space, $R^M(s, a)$ is the reward distibution over $\mathbb{R}$ in state $s$ with action $a$, $P^M(\cdot|s, a)$ is the transition probability over $\mathcal{S}$ from state $s$ with action $a$, $\tau$ is the time horizon, and $\rho$ the initial state distribution. We define the MDP and all other random variables we will consider with respect to a probability space $(\Omega, \mathcal{F}, \mathbb{P})$.

A deterministic policy $\mu$ is a function mapping each state $s \in \mathcal{S}$ and $i = 1, \ldots, \tau$ to an action $a \in \mathcal{A}$. For each MDP $M = (\mathcal{S}, \mathcal{A}, R^M, P^M, \tau, \rho)$ and policy $\mu$, we define a value function

$$V_{\mu,i}^M(s) := \mathbb{E}_{M,\mu} \left[ \sum_{j=i}^{\tau} \overline{R}^M(s_j, a_j) \Big| s_i = s \right],$$

where $\overline{R}^M(s, a)$ denotes the expected reward realized when action $a$ is selected while in state $s$, and the subscripts of the expectation operator indicate that $a_j = \mu(s_j, j)$, and $s_{j+1} \sim P^M(\cdot|s_j, a_j)$ for $j = i, \ldots, \tau$. A policy $\mu$ is optimal for the MDP $M$ if $V_{\mu,i}^M(s) = \max_{\mu'} V_{\mu',i}^M(s)$ for all $s \in \mathcal{S}$ and $i = 1, \ldots, \tau$. We will associate with each MDP $M$ a policy $\mu^M$ that is optimal for $M$.

The reinforcement learning agent interacts with the MDP over episodes that begin at times $t_k = (k-1)\tau + 1$, $k = 1, 2, \ldots$. At each time $t$, the agent selects an action $a_t$, observes a scalar reward $r_t$, and then transitions to $s_{t+1}$. Let $H_t = (s_1, a_1, r_1, \ldots, s_{t-1}, a_{t-1}, r_{t-1})$ denote the history of observations made *prior* to time $t$. A reinforcement learning algorithm is a deterministic sequence $\{\pi_k | k = 1, 2, \ldots\}$ of functions, each mapping $H_{t_k}$ to a probability distribution $\pi_k(H_{t_k})$ over policies which the agent will employ during the $k$th episode. We define the regret incurred by a reinforcement learning algorithm $\pi$ up to time $T$ to be:

$$\text{Regret}(T, \pi, M^*) := \sum_{k=1}^{\lceil T/\tau \rceil} \Delta_k,$$

where $\Delta_k$ denotes regret over the $k$th episode, defined with respect to the MDP $M^*$ by

$$\Delta_k := \sum_{\mathcal{S}} \rho(s)(V_{\mu^*,1}^{M^*}(s) - V_{\mu_k,1}^{M^*}(s))$$

with $\mu^* = \mu^{M^*}$ and $\mu_k \sim \pi_k(H_{t_k})$. Note that regret is not deterministic since it can depend on the random MDP $M^*$, the algorithm's internal random sampling and, through the history $H_{t_k}$, on previous random transitions and random rewards. We will assess and compare algorithm performance in terms of regret and its expectation.

## 3 Factored MDPs

Intuitively a factored MDP is an MDP whose rewards and transitions exhibit some conditional independence structure. To formalize this definition we must introduce some more notation common to the literature [11].

**Definition 1** (Scope operation for factored sets $\mathcal{X} = \mathcal{X}_1 \times .. \times \mathcal{X}_n$).
*For any subset of indices $Z \subseteq \{1, 2, .., n\}$ let us define the scope set $\mathcal{X}[Z] := \bigotimes_{i \in Z} \mathcal{X}_i$. Further, for any $x \in \mathcal{X}$ define the scope variable $x[Z] \in \mathcal{X}[Z]$ to be the value of the variables $x_i \in \mathcal{X}_i$ with indices $i \in Z$. For singleton sets $Z$ we will write $x[i]$ for $x[\{i\}]$ in the natural way.*

Let $\mathcal{P}_{\mathcal{X}, \mathcal{Y}}$ be the set of functions mapping elements of a finite set $\mathcal{X}$ to probability mass functions over a finite set $\mathcal{Y}$. $\mathcal{P}_{\mathcal{X}, \mathbb{R}}^{C, \sigma}$ will denote the set of functions mapping elements of a finite set $\mathcal{X}$ to $\sigma$-sub-Gaussian probability measures over $(\mathbb{R}, \mathcal{B}(\mathbb{R}))$ with mean bounded in $[0, C]$. For reinforcement learning we will write $\mathcal{X}$ for $\mathcal{S} \times \mathcal{A}$ and consider factored reward and factored transition functions which are drawn from within these families.

**Definition 2** ( Factored reward functions $R \in \mathcal{R} \subseteq \mathcal{P}_{\mathcal{X}, \mathbb{R}}^{C; \sigma}$).
*The reward function class $\mathcal{R}$ is factored over $\mathcal{S} \times \mathcal{A} = \mathcal{X} = \mathcal{X}_1 \times .. \times \mathcal{X}_n$ with scopes $Z_1, ..Z_l$ if and only if, for all $R \in \mathcal{R}, x \in \mathcal{X}$ there exist functions $\{R_i \in \mathcal{P}_{\mathcal{X}[Z_i], \mathbb{R}}^{C, \sigma}\}_{i=1}^l$ such that,*

$$\mathbb{E}[r] = \sum_{i=1}^{l} \mathbb{E}[r_i]$$

*for $r \sim R(x)$ is equal to $\sum_{i=1}^l r_i$ with each $r_i \sim R_i(x[Z_i])$ and individually observed.*

**Definition 3** ( Factored transition functions $P \in \mathcal{P} \subseteq \mathcal{P}_{\mathcal{X}, \mathcal{S}}$ ).
*The transition function class $\mathcal{P}$ is factored over $\mathcal{S} \times \mathcal{A} = \mathcal{X} = \mathcal{X}_1 \times .. \times \mathcal{X}_n$ and $\mathcal{S} = \mathcal{S}_1 \times .. \times \mathcal{S}_m$ with scopes $Z_1, ..Z_m$ if and only if, for all $P \in \mathcal{P}, x \in \mathcal{X}, s \in \mathcal{S}$ there exist some $\{P_i \in \mathcal{P}_{\mathcal{X}[Z_i], \mathcal{S}_i}\}_{i=1}^m$ such that,*

$$P(s|x) = \prod_{i=1}^{m} P_i \left( s[i] \,\middle|\, x[Z_i] \right)$$

A factored MDP (FMDP) is then defined to be an MDP with both factored rewards and factored transitions. Writing $\mathcal{X} = \mathcal{S} \times \mathcal{A}$ a FMDP is fully characterized by the tuple

$$M = \left( \{\mathcal{S}_i\}_{i=1}^m; \; \{\mathcal{X}_i\}_{i=1}^n; \; \{Z_i^R\}_{i=1}^l; \; \{R_i\}_{i=1}^l; \; \{Z_i^P\}_{i=1}^m; \; \{P_i\}_{i=1}^m; \; \tau; \; \rho \right),$$

where $Z_i^R$ and $Z_i^P$ are the scopes for the reward and transition functions respectively in $\{1, .., n\}$ for $\mathcal{X}_i$. We assume that the size of all scopes $|Z_i| \leq \zeta \ll n$ and factors $|\mathcal{X}_i| \leq K$ so that the domains of $R_i$ and $P_i$ are of size at most $K^\zeta$.

## 4 Results

Our first result shows that we can bound the expected regret of PSRL.

**Theorem 1** (Expected regret for PSRL in factored MDPs).
*Let $M^*$ be factored with graph structure $\mathcal{G} = \left( \{\mathcal{S}_i\}_{i=1}^m; \; \{\mathcal{X}_i\}_{i=1}^n; \; \{Z_i^R\}_{i=1}^l; \; \{Z_i^P\}_{i=1}^m; \; \tau \right)$. If $\phi$ is the distribution of $M^*$ and $\Psi$ is the span of the optimal value function then we can bound the regret of PSRL:*

$$\mathbb{E}\left[ \mathrm{Regret}(T, \pi_\tau^{\mathrm{PS}}, M^*) \right] \leq \sum_{i=1}^{l} \left\{ 5\tau C |\mathcal{X}[Z_i^R]| + 12\sigma \sqrt{|\mathcal{X}[Z_i^R]| T \log \left( 4l |\mathcal{X}[Z_i^R]| kT \right)} \right\} + 2\sqrt{T}$$

$$+ 4 + \mathbb{E}[\Psi] \left( 1 + \frac{4}{T-4} \right) \sum_{j=1}^{m} \left\{ 5\tau |\mathcal{X}[Z_j^P]| + 12\sqrt{|\mathcal{X}[Z_j^P]| |\mathcal{S}_j| T \log \left( 4m |\mathcal{X}[Z_j^P]| kT \right)} \right\} \quad (1)$$

We have a similar result for UCRL-Factored that holds with high probability.

**Theorem 2** (High probability regret for UCRL-Factored in factored MDPs)**.**
*Let $M^*$ be factored with graph structure $\mathcal{G} = \left(\{\mathcal{S}_i\}_{i=1}^m; \{\mathcal{X}_i\}_{i=1}^n; \{Z_i^R\}_{i=1}^l; \{Z_i^P\}_{i=1}^m; \tau\right)$. If $D$ is the diameter of $M^*$, then for any $M^*$ can bound the regret of UCRL-Factored:*

$$\text{Regret}(T, \pi_\tau^{\text{UC}}, M^*) \leq \sum_{i=1}^{l} \left\{ 5\tau C|\mathcal{X}[Z_i^R]| + 12\sigma\sqrt{|\mathcal{X}[Z_i^R]|T\log\left(12l|\mathcal{X}[Z_i^R]|kT/\delta\right)} \right\} + 2\sqrt{T}$$

$$+ CD\sqrt{2T\log(6/\delta)} + CD\sum_{j=1}^{m} \left\{ 5\tau|\mathcal{X}[Z_j^P]| + 12\sqrt{|\mathcal{X}[Z_j^P]||\mathcal{S}_j|T\log\left(12m|\mathcal{X}[Z_j^P]|kT/\delta\right)} \right\} (2)$$

*with probability at least $1 - \delta$*

Both algorithms give bounds $\tilde{O}\left(\Xi\sum_{j=1}^{m}\sqrt{|\mathcal{X}[Z_j^P]||S_j|T}\right)$ where $\Xi$ is a measure of MDP connectedness: expected span $\mathbb{E}[\Psi]$ for PSRL and scaled diameter $CD$ for UCRL-Factored. The span of an MDP is the maximum difference in value of any two states under the optimal policy $\Psi(M^*) := \max_{s,s' \in \mathcal{S}}\{V_{\mu^*,1}^{M^*}(s) - V_{\mu^*,1}^{M^*}(s')\}$. The diameter of an MDP is the maximum number of expected timesteps to get between any two states $D(M^*) = \max_{s \neq s'} \min_\mu T_{s \to s'}^\mu$. PSRL's bounds are tighter since $\Psi(M) \leq CD(M)$ and may be exponentially smaller.

However, UCRL-Factored has stronger probabilistic guarantees than PSRL since its bounds hold with high probability for any MDP $M^*$ not just in expectation. There is an optimistic algorithm REGAL [5] which formally replaces the UCRL2 $D$ with $\Psi$ and retains the high probability guarantees. An analogous extension to REGAL-Factored is possible, however, no practical implementation of that algorithm exists even with an FMDP planner.

The algebra in Theorems 1 and 2 can be overwhelming. For clarity, we present a symmetric problem instance for which we can produce a cleaner single-term upper bound. Let $\mathcal{Q}$ be shorthand for the simple graph structure with $l + 1 = m$, $C = \sigma = 1$, $|\mathcal{S}_i| = |\mathcal{X}_i| = K$ and $|Z_i^R| = |Z_j^P| = \zeta$ for $i = 1,..,l$ and $j = 1,..,m$, we will write $J = K^\zeta$.

**Corollary 1** (Clean bounds for PSRL in a symmetric problem)**.**
*If $\phi$ is the distribution of $M^*$ with structure $\mathcal{Q}$ then we can bound the regret of PSRL:*

$$\mathbb{E}\left[\text{Regret}(T, \pi_\tau^{\text{PS}}, M^*)\right] \leq 15m\tau\sqrt{JKT\log(2mJT)} \tag{3}$$

**Corollary 2** (Clean bounds for UCRL-Factored in a symmetric problem)**.**
*For any MDP $M^*$ with structure $\mathcal{Q}$ we can bound the regret of UCRL-Factored:*

$$\text{Regret}(T, \pi_\tau^{\text{UC}}, M^*) \leq 15m\tau\sqrt{JKT\log(12mJT/\delta)} \tag{4}$$

*with probability at least $1 - \delta$.*

Both algorithms satisfy bounds of $\tilde{O}(\tau m\sqrt{JKT})$ which is exponentially tighter than can be obtained by any $\mathcal{Q}$-naive algorithm. For a factored MDP with $m$ independent components with $S$ states and $A$ actions the bound $\tilde{O}(mS\sqrt{AT})$ is close to the lower bound $\Omega(m\sqrt{SAT})$ and so the bound is near optimal. The corollaries follow directly from Theorems 1 and 2 as shown in Appendix B.

## 5   Confidence sets

Our analysis will rely upon the construction of confidence sets based around the empirical estimates for the underlying reward and transition functions. The confidence sets are constructed to contain the true MDP with high probability. This technique is common to the literature, but we will exploit the additional graph structure $\mathcal{G}$ to sharpen the bounds.

Consider a family of functions $\mathcal{F} \subseteq \mathcal{M}_{\mathcal{X},(\mathcal{Y},\Sigma_\mathcal{Y})}$ which takes $x \in \mathcal{X}$ to a probability distribution over $(\mathcal{Y},\Sigma_\mathcal{Y})$. We will write $\mathcal{M}_{\mathcal{X},\mathcal{Y}}$ unless we wish to stress a particular $\sigma$-algebra.
**Definition 4** (Set widths)**.**
*Let $\mathcal{X}$ be a finite set, and let $(\mathcal{Y},\Sigma_\mathcal{Y})$ be a measurable space. The width of a set $\mathcal{F} \in \mathcal{M}_{\mathcal{X},\mathcal{Y}}$ at $x \in \mathcal{X}$ with respect to a norm $\|\cdot\|$ is*

$$w_\mathcal{F}(x) := \sup_{\overline{f},\underline{f} \in \mathcal{F}} \|(\overline{f} - \underline{f})(x)\|$$

Our confidence set sequence $\{\mathcal{F}_t \subseteq \mathcal{F} : t \in \mathbb{N}\}$ is initialized with a set $\mathcal{F}$. We adapt our confidence set to the observations $y_t \in \mathcal{Y}$ which are drawn from the true function $f^* \in \mathcal{F}$ at measurement points $x_t \in \mathcal{X}$ so that $y_t \sim f^*(x_t)$. Each confidence set is then centered around an empirical estimate $\hat{f}_t \in \mathcal{M}_{\mathcal{X},\mathcal{Y}}$ at time $t$, defined by

$$\hat{f}_t(x) = \frac{1}{n_t(x)} \sum_{\tau < t : x_\tau = x} \delta_{y_\tau},$$

where $n_t(x)$ is the number of time $x$ appears in $(x_1, .., x_{t-1})$ and $\delta_{y_t}$ is the probability mass function over $\mathcal{Y}$ that assigns all probability to the outcome $y_t$.

Our sequence of confidence sets depends on our choice of norm $\| \cdot \|$ and a non-decreasing sequence $\{d_t : t \in \mathbb{N}\}$. For each $t$, the confidence set is defined by:

$$\mathcal{F}_t = \mathcal{F}_t(\| \cdot \|, x_1^{t-1}, d_t) := \left\{ f \in \mathcal{F} \;\middle|\; \|(f - \hat{f}_t)(x_i)\| \leq \sqrt{\frac{d_t}{n_t(x_i)}} \; \forall i = 1, .., t-1 \right\}.$$

Where $x_1^{t-1}$ is shorthand for $(x_1, .., x_{t-1})$ and we interpret $n_t(x_i) = 0$ as a null constraint. The following result shows that we can bound the sum of confidence widths through time.

**Theorem 3** (Bounding the sum of widths)**.**
*For all finite sets $\mathcal{X}$, measurable spaces $(\mathcal{Y}, \Sigma_\mathcal{Y})$, function classes $\mathcal{F} \subseteq \mathcal{M}_{\mathcal{X},\mathcal{Y}}$ with uniformly bounded widths $w_\mathcal{F}(x) \leq C_\mathcal{F} \; \forall x \in \mathcal{X}$ and non-decreasing sequences $\{d_t : t \in \mathbb{N}\}$:*

$$\sum_{k=1}^{L} \sum_{i=1}^{\tau} w_{\mathcal{F}_k}(x_{t_k+i}) \leq 4\big(\tau C_\mathcal{F} |\mathcal{X}| + 1\big) + 4\sqrt{2 d_T |\mathcal{X}| T} \tag{5}$$

*Proof.* The proof follows from elementary counting arguments on $n_t(x)$ and the pigeonhole principle. A full derivation is given in Appendix A. $\qquad\square$

## 6 Algorithms

With our notation established, we are now able to introduce our algorithms for efficient learning in Factored MDPs. PSRL and UCRL-Factored proceed in episodes of fixed policies. At the start of the $k$th episode they produce a candidate MDP $M_k$ and then proceed with the policy which is optimal for $M_k$. In PSRL, $M_k$ is generated by a sample from the posterior for $M^*$, whereas UCRL-Factored chooses $M_k$ optimistically from the confidence set $\mathcal{M}_k$.

Both algorithms require prior knowledge of the graphical structure $\mathcal{G}$ and an approximate planner for FMDPs. We will write $\Gamma(M, \epsilon)$ for a planner which returns $\epsilon$-optimal policy for $M$. We will write $\tilde{\Gamma}(\mathcal{M}, \epsilon)$ for a planner which returns an $\epsilon$-optimal policy for most optimistic realization from a family of MDPs $\mathcal{M}$. Given $\Gamma$ it is possible to obtain $\tilde{\Gamma}$ through extended value iteration, although this might become computationally intractable [4].

PSRL remains identical to earlier treatment [13, 6] provided $\mathcal{G}$ is encoded in the prior $\phi$. UCRL-Factored is a modification to UCRL2 that can exploit the graph and episodic structure of . We write $\mathcal{R}_t^i(d_t^{R_i})$ and $\mathcal{P}_t^j(d_t^{P_j})$ as shorthand for these confidence sets $\mathcal{R}_t^i(|\mathbb{E}[\cdot]|, x_1^{t-1}[Z_i^R], d_t^{R_i})$ and $\mathcal{P}_t^i(\| \cdot \|_1, x_1^{t-1}[Z_j^P], d_t^{P_j})$ generated from initial sets $\mathcal{R}_1^i = \mathcal{P}_{\mathcal{X}[Z_i^R],\mathbb{R}}^{C,\sigma}$ and $\mathcal{P}_1^j = \mathcal{P}_{\mathcal{X}[Z_j^P],\mathcal{S}_j}$.

We should note that UCRL2 was designed to obtain regret bounds even in MDPs without episodic reset. This is accomplished by imposing artificial episodes which end whenever the number of visits to a state-action pair is doubled [4]. It is simple to extend UCRL-Factored's guarantees to this setting using this same strategy. This will not work for PSRL since our current analysis requires that the episode length is independent of the sampled MDP. Nevertheless, there has been good empirical performance using this method for MDPs without episodic reset in simulation [6].

| **Algorithm 1** | **Algorithm 2** |
|---|---|
| PSRL (Posterior Sampling) | UCRL-Factored (Optimism) |

| **Algorithm 1** PSRL (Posterior Sampling) |
|---|
| 1: **Input:** Prior $\phi$ encoding $\mathcal{G}$, $t = 1$ |
| 2: **for** episodes $k = 1, 2, ..$ **do** |
| 3: $\quad$ sample $M_k \sim \phi(\cdot\|H_t)$ |
| 4: $\quad$ compute $\mu_k = \Gamma(M_k, \sqrt{\tau/k})$ |
| 5: $\quad$ **for** timesteps $j = 1, .., \tau$ **do** |
| 6: $\quad\quad$ sample and apply $a_t = \mu_k(s_t, j)$ |
| 7: $\quad\quad$ observe $r_t$ and $s_{t+1}^m$ |
| 8: $\quad\quad$ $t = t + 1$ |
| 9: $\quad$ **end for** |
| 10: **end for** |

| **Algorithm 2** UCRL-Factored (Optimism) |
|---|
| 1: **Input:** Graph structure $\mathcal{G}$, confidence $\delta$, $t = 1$ |
| 2: **for** episodes $k = 1, 2, ..$ **do** |
| 3: $\quad$ $d_t^{R_i} = 4\sigma^2 \log\left(4l\|\mathcal{X}[Z_i^R]\|k/\delta\right)$ for $i = 1, .., l$ |
| 4: $\quad$ $d_t^{P_j} = 4\|\mathcal{S}_j\| \log\left(4m\|\mathcal{X}[Z_j^P]\|k/\delta\right)$ for $j = 1, .., m$ |
| 5: $\quad$ $\mathcal{M}_k = \{M \mid \mathcal{G}, \overline{R}_i \in \mathcal{R}_t^i(d_t^{R_i}), P_j \in \mathcal{P}_t^j(d_t^{P_j}) \,\forall i, j\}$ |
| 6: $\quad$ compute $\mu_k = \tilde{\Gamma}(\mathcal{M}_k, \sqrt{\tau/k})$ |
| 7: $\quad$ **for** timesteps $u = 1, .., \tau$ **do** |
| 8: $\quad\quad$ sample and apply $a_t = \mu_k(s_t, u)$ |
| 9: $\quad\quad$ observe $r_t^1, .., r_t^l$ and $s_{t+1}^1, .., s_{t+1}^m$ |
| 10: $\quad\quad$ $t = t + 1$ |
| 11: $\quad$ **end for** |
| 12: **end for** |

# 7 Analysis

For our common analysis of PSRL and UCRL-Factored we will let $\tilde{M}_k$ refer generally to either the sampled MDP used in PSRL or the optimistic MDP chosen from $\mathcal{M}_k$ with associated policy $\tilde{\mu}_k$). We introduce the Bellman operator $\mathcal{T}_\mu^M$, which for any MDP $M = (\mathcal{S}, \mathcal{A}, R^M, P^M, \tau, \rho)$, stationary policy $\mu : \mathcal{S} \to \mathcal{A}$ and value function $V : \mathcal{S} \to \mathbb{R}$, is defined by

$$\mathcal{T}_\mu^M V(s) := \overline{R}^M(s, \mu(s)) + \sum_{s' \in \mathcal{S}} P^M(s'\|s, \mu(s))V(s').$$

This returns the expected value of state $s$ where we follow the policy $\mu$ under the laws of $M$, for one time step. We will streamline our discussion of $P^M, R^M, V_{\mu,i}^M$ and $\mathcal{T}_\mu^M$ by simply writing $*$ in place of $M^*$ or $\mu^*$ and $k$ in place of $\tilde{M}_k$ or $\tilde{\mu}_k$ where appropriate; for example $V_{k,i}^* := V_{\tilde{\mu}_k,i}^{M^*}$. We will also write $x_{k,i} := (s_{t_k+i}, \mu_k(s_{t_k+i}))$.

We now break down the regret by adding and subtracting the *imagined* near optimal reward of policy $\tilde{\mu}_K$, which is known to the agent. For clarity of analysis we consider only the case of $\rho(s') = \mathbb{1}\{s' = s\}$ but this changes nothing for our consideration of finite $\mathcal{S}$.

$$\Delta_k = V_{*,1}^*(s) - V_{k,1}^*(s) = \left(V_{k,1}^k(s) - V_{k,1}^*(s)\right) + \left(V_{*,1}^*(s) - V_{k,1}^k(s)\right) \tag{6}$$

$V_{*,1}^* - V_{k,1}^k$ relates the optimal rewards of the MDP $M^*$ to those near optimal for $\tilde{M}_k$. We can bound this difference by the planning accuracy $\sqrt{1/k}$ for PSRL in expectation, since $M^*$ and $M_k$ are equal in law, and for UCRL-Factored in high probability by optimism.

We decompose the first term through repeated application of dynamic programming:

$$\left(V_{k,1}^k - V_{k,1}^*\right)(s_{t_k+1}) = \sum_{i=1}^{\tau} \left(\mathcal{T}_{k,i}^k - \mathcal{T}_{k,i}^*\right) V_{k,i+1}^k(s_{t_k+i}) + \sum_{i=1}^{\tau} d_{t_k+1}. \tag{7}$$

Where $d_{t_k+i} := \sum_{s \in \mathcal{S}} \left\{P^*(s\|x_{k,i})(V_{k,i+1}^* - V_{k,i+1}^k)(s)\right\} - (V_{k,i+1}^* - V_{k,i+1}^k)(s_{t_k+i})$ is a martingale difference bounded by $\Psi_k$, the span of $V_{k,i}^k$. For UCRL-Factored we can use optimism to say that $\Psi_k \leq CD$ [4] and apply the Azuma-Hoeffding inequality to say that:

$$\mathbb{P}\left(\sum_{k=1}^{m} \sum_{i=1}^{\tau} d_{t_k+i} > CD\sqrt{2T\log(2/\delta)}\right) \leq \delta \tag{8}$$

The remaining term is the one step Bellman error of the imagined MDP $\tilde{M}_k$. Crucially this term only depends on states and actions $x_{k,i}$ which are actually observed. We can now use

the Hölder inequality to bound

$$\sum_{i=1}^{\tau} \left(\mathcal{T}_{k,i}^k - \mathcal{T}_{k,i}^*\right) V_{k,i+1}^k(s_{t_k+i}) \le \sum_{i=1}^{\tau} |\overline{R}^k(x_{k,i}) - \overline{R}^*(x_{k,i})| + \frac{1}{2}\Psi_k \|P^k(\cdot|x_{k,i}) - P^*(\cdot|x_{k,i})\|_1 \quad (9)$$

## 7.1 Factorization decomposition

We aim to exploit the graphical structure $\mathcal{G}$ to create more efficient confidence sets $\mathcal{M}_k$. It is clear from (9) that we may upper bound the deviations of $\overline{R}^*, \overline{R}^k$ factor-by-factor using the triangle inequality. Our next result, Lemma 1, shows we can also do this for the transition functions $P^*$ and $P^k$. This is the key result that allows us to build confidence sets around each factor $P_j^*$ rather than $P^*$ as a whole.

**Lemma 1** (Bounding factored deviations).
*Let the transition function class $\mathcal{P} \subseteq \mathcal{P}_{\mathcal{X},\mathcal{S}}$ be factored over $\mathcal{X} = \mathcal{X}_1 \times .. \times \mathcal{X}_n$ and $\mathcal{S} = \mathcal{S}_1 \times .. \times \mathcal{S}_m$ with scopes $Z_1, ..Z_m$. Then, for any $P, \tilde{P} \in \mathcal{P}$ we may bound their L1 distance by the sum of the differences of their factorizations:*

$$\|P(x) - \tilde{P}(x)\|_1 \le \sum_{i=1}^{m} \|P_i(x[Z_i]) - \tilde{P}_i(x[Z_i])\|_1$$

*Proof.* We begin with the simple claim that for any $\alpha_1, \alpha_2, \beta_1, \beta_2 \in (0,1]$:

$$\begin{aligned}
|\alpha_1\alpha_2 - \beta_1\beta_2| &= \alpha_2 \left|\alpha_1 - \frac{\beta_1\beta_2}{\alpha_2}\right| \\
&\le \alpha_2 \left(|\alpha_1 - \beta_1| + \left|\beta_1 - \frac{\beta_1\beta_2}{\alpha_2}\right|\right) \\
&\le \alpha_2|\alpha_1 - \beta_1| + \beta_1|\alpha_2 - \beta_2|
\end{aligned}$$

This result also holds for any $\alpha_1, \alpha_2, \beta_1, \beta_2 \in [0,1]$, where 0 can be verified case by case.

We now consider the probability distributions $p, \tilde{p}$ over $\{1, .., d_1\}$ and $q, \tilde{q}$ over $\{1, .., d_2\}$. We let $Q = pq^T, \tilde{Q} = \tilde{p}\tilde{q}^T$ be the joint probability distribution over $\{1, .., d_1\} \times \{1, .., d_2\}$. Using the claim above we bound the L1 deviation $\|Q - \tilde{Q}\|_1$ by the deviations of their factors:

$$\begin{aligned}
\|Q - \tilde{Q}\|_1 &= \sum_{i=1}^{d_1}\sum_{j=1}^{d_2} |p_i q_j - \tilde{p}_i \tilde{q}_j| \\
&\le \sum_{i=1}^{d_1}\sum_{j=1}^{d_2} q_j|p_i - \tilde{p}_i| + \tilde{p}_i|q_j - \tilde{q}_j| \\
&= \|p - \tilde{p}\|_1 + \|q - \tilde{q}\|_1
\end{aligned}$$

We conclude the proof by applying this $m$ times to the factored transitions $P$ and $\tilde{P}$. $\square$

## 7.2 Concentration guarantees for $\mathcal{M}_k$

We now want to show that the true MDP lies within $\mathcal{M}_k$ with high probability. Note that posterior sampling will also allow us to then say that the sampled $M_k$ is within $\mathcal{M}_k$ with high probability too. In order to show this, we first present a concentration result for the L1 deviation of empirical probabilities.

**Lemma 2** (L1 bounds for the empirical transition function).
*For all finite sets $\mathcal{X}$, finite sets $\mathcal{Y}$, function classes $\mathcal{P} \subseteq \mathcal{P}_{\mathcal{X},\mathcal{Y}}$ then for any $x \in \mathcal{X}$, $\epsilon > 0$ the deviation the true distribution $P^*$ to the empirical estimate after $t$ samples $\hat{P}_t$ is bounded:*

$$\mathbb{P}\left(\|P^*(x) - \hat{P}_t(x)\|_1 \ge \epsilon\right) \le \exp\left(|\mathcal{Y}|\log(2) - \frac{n_t(x)\epsilon^2}{2}\right)$$

*Proof.* This is a relaxation of the result proved by Weissman [19]. □

Lemma 2 ensures that for any $x \in \mathcal{X}$ $\mathbb{P}(\|P_j^*(x) - \hat{P}_{j_t}(x)\|_1 \geq \sqrt{\frac{2|\mathcal{S}_j|}{n_t(x)} \log\left(\frac{2}{\delta'}\right)}) \leq \delta'$. We then define $d_{t_k}^{P_j} = 2|\mathcal{S}_i| \log(2/\delta'_{k,j})$ with $\delta'_{k,j} = \delta/(2m|\mathcal{X}[Z_j^P]|k^2)$. Now using a union bound we conclude $\mathbb{P}(P_j^* \in \mathcal{P}_t^j(d_{t_k}^{P_j}) \; \forall k \in \mathbb{N}, j = 1,..,m) \geq 1 - \delta$.

**Lemma 3** (Tail bounds for sub $\sigma$-gaussian random variables)**.**
*If $\{\epsilon_i\}$ are all independent and sub $\sigma$-gaussian then $\forall \beta \geq 0$:*

$$\mathbb{P}\left(\frac{1}{n}|\sum_{i=1}^n \epsilon_i| > \beta\right) \leq \exp\left(\log(2) - \frac{n\beta^2}{2\sigma^2}\right)$$

A similar argument now ensures that $\mathbb{P}\left(\overline{R}_i^* \in \mathcal{R}_t^i(d_{t_k}^{R_i}) \; \forall k \in \mathbb{N}, i = 1,..,l\right) \geq 1 - \delta$, and so

$$\mathbb{P}\left(M^* \in \mathcal{M}_k \; \forall k \in \mathbb{N}\right) \geq 1 - 2\delta \quad (10)$$

### 7.3   Regret bounds

We now have all the necessary intermediate results to complete our proof. We begin with the analysis of PSRL. Using equation (10) and the fact that $M^*, M_k$ are equal in law by posterior sampling, we can say that $\mathbb{P}(M^*, M_k \in \mathcal{M}_k \forall k \in \mathbb{N}) \geq 1 - 4\delta$. The contributions from regret in planning function $\Gamma$ are bounded by $\sum_{k=1}^m \sqrt{\tau/k} \leq 2\sqrt{T}$. From here we take equation (9), Lemma 1 and Theorem 3 to say that for any $\delta > 0$:

$$\mathbb{E}\left[\text{Regret}(T, \pi_\tau^{\text{PS}}, M^*)\right] \quad \leq \quad 4\delta T + 2\sqrt{T} + \sum_{i=1}^l \left\{4(\tau C|\mathcal{X}[Z_i^R]| + 1) + 4\sqrt{2d_T^{R_i}|\mathcal{X}[Z_i^R]|T}\right\}$$

$$+ \sup_{k=1,..,L} \left(\mathbb{E}[\Psi_k | M_k, M^* \in \mathcal{M}_k]\right) \times \sum_{j=1}^m \left\{4(\tau|\mathcal{X}[Z_j^P]| + 1) + 4\sqrt{2d_T^{P_j}|\mathcal{X}[Z_j^P]|T}\right\}$$

Let $A = \{M^*, M_k \in \mathcal{M}_k\}$, since $\Psi_k \geq 0$ and by posterior sampling $\mathbb{E}[\Psi_k] = \mathbb{E}[\Psi]$ for all $k$:

$$\mathbb{E}[\Psi_k | A] \leq \mathbb{P}(A)^{-1} \mathbb{E}[\Psi] \leq \left(1 - \frac{4\delta}{k^2}\right)^{-1} \mathbb{E}[\Psi] = \left(1 + \frac{4\delta}{k^2 - 4\delta}\right) \mathbb{E}[\Psi] \leq \left(1 + \frac{4\delta}{1 - 4\delta}\right) \mathbb{E}[\Psi].$$

Plugging in $d_T^{R_i}$ and $d_T^{P_j}$ and setting $\delta = 1/T$ completes the proof of Theorem 1. The analysis of UCRL-Factored and Theorem 2 follows similarly from (8) and (10). Corollaries 1 and 2 follow from substituting the structure $\mathcal{Q}$ and upper bounding the constant and logarithmic terms. This is presented in detail in Appendix B.

## 8   Conclusion

We present the first algorithms with near-optimal regret bounds in factored MDPs. Many practical problems for reinforcement learning will have extremely large state and action spaces, this allows us to obtain meaningful performance guarantees even in previously intractably large systems. However, our analysis leaves several important questions unaddressed. First, we assume access to an approximate FMDP planner that may be computationally prohibitive in practice. Second, we assume that the graph structure is known a priori but there are other algorithms that seek to learn this from experience [20, 21]. Finally, we might consider dimensionality reduction in large MDPs more generally, where either the rewards, transitions or optimal value function are known to belong in some function class $\mathcal{F}$ to obtain bounds that depend on the dimensionality of $\mathcal{F}$.

### Acknowledgments

Osband is supported by Stanford Graduate Fellowships courtesy of PACCAR inc. This work was supported in part by Award CMMI-0968707 from the National Science Foundation.

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
