[Supplementary Material]

# A  Bounding the widths of confidence sets

We present elementary arguments which culminate in a proof of Theorem 3.

**Lemma 4** (Concentration results for $\sqrt{d_T/n_t(x)}$)**.**
*For all finite sets $\mathcal{X}$ and any $d_T, \epsilon \geq 0$:*

$$\sum_{t=1}^{T} \mathbb{1}\left\{\sqrt{d_T/n_t(x_t)} > h(d_T, \epsilon)\right\} \leq \sum_{t=1}^{T} \mathbb{1}\left\{\sqrt{d_T/n_t(x_t)} > \epsilon\right\} + |\mathcal{X}|,$$

*Where $h(d_T, \epsilon) := \sqrt{d_T \epsilon^2/(d_T + \epsilon^2)}$.*

*Proof.* Let $(x_{s_1}, .., x_{s_K})$ be the largest subsequence of $x_1^T$ such that $\sqrt{d_T/n_{s_i}(x_{s_i})} \in (h(d_T, \epsilon), \epsilon] \; \forall i$. Now for any $x \in \mathcal{X}$, let $\mathcal{T}_x = \{s_i \mid x_{s_i} = x\}$. Suppose there exist two distinct elements $\sigma, \rho \in \mathcal{T}_x$ with $\sigma < \rho$ so that $n_\rho(x) \geq n_\sigma(x) + 1$. We note that for any $n \in \mathbb{R}_+$, $h(d_T, \sqrt{d_T/n}) = \sqrt{d_T/(n+1)}$ so that:

$$\epsilon \geq \sqrt{d_T/n_\sigma(x)} \implies h(d_T, \epsilon) \geq \sqrt{d_T/(n_\sigma(x)+1)} \geq \sqrt{d_T/n_\rho(x)}$$

This contradicts our assumption $\sqrt{d_T/n_\rho(x)} \in (h(d, \epsilon), \epsilon]$ and so we must conclude that $|\mathcal{T}_x| \leq 1$ for all $x \in \mathcal{X}$. This means that $(x_{s_1}, .., x_{s_K})$ forms a subsequence of unique elements in $\mathcal{X}$, the total length of which must be bounded by $|\mathcal{X}|$. □

We now provide a corollary of this result which allows for episodic delays in updating visit counts $n_t(x)$. We imagine that we will only update our counts every $\tau$ steps.

**Corollary 3** (Concentration results for $\sqrt{d_T/n_{t_k}(x)}$ in the episodic setting)**.**
*Let us associate times within episodes of length $\tau$, $t = t_k + i$ for $i = 1, .., \tau$ and $T = M \times \tau$. For all finite sets $\mathcal{X}$ and any $d_T, \epsilon \geq 0$:*

$$\sum_{k=1}^{M} \sum_{i=1}^{\tau} \mathbb{1}\left\{\sqrt{d_T/n_{t_k}(x_{t_k+i})} > h^{(\tau)}(d_T, \epsilon)\right\} \leq \sum_{k=1}^{M} \sum_{i=1}^{\tau} \mathbb{1}\left\{\sqrt{d_T/n_{t_k}(x_{t_k+i})} > \epsilon\right\} + 2\tau|\mathcal{X}|,$$

*Where $h^{(\tau)}(d_T, \epsilon)$ is the $\tau$-fold composition of $h(d_T, \cdot)$ acting on $\epsilon$.*

*Proof.* By an argument of visiting times similar to lemma 4 we can see that the worst case scenario for the episodic case $\sum_{k=1}^{M} \sum_{i=1}^{\tau} \mathbb{1}\left\{\sqrt{d_T/n_{t_k}(x_{t_k+i})} > h^{(\tau)}(d_T, \epsilon)\right\}$ is to visit each $x$ exactly $\tau - 1$ times before the start of an episode, and then spend the entirety of the following episode within the state. Here we have upper bounded $2\tau - 1$ by $2\tau$ and $|\mathcal{X}| - 1$ by $|\mathcal{X}|$ to complete our result. □

It will be useful to define notion of radius for each confidence set at each $x \in \mathcal{X}$, $r_{\mathcal{F}_t}(x) := \sup_{f \in \mathcal{F}_t} \|(f - \hat{f}_t)(x)\|$. By the triangle inequality, we have $w_{\mathcal{F}_t}(x) \leq 2 r_{\mathcal{F}_t}(x)$ for all $x \in \mathcal{X}$.

**Lemma 5** (Bounding the number of large radii)**.**
*Let us write $\mathcal{F}_k$ for $\mathcal{F}_{t_k}$ and associate times within episodes of length $\tau$, $t = t_k + i$ for $i = 1, .., \tau$ and $T = M \times \tau$. For all finite sets $\mathcal{X}$, measurable spaces $(\mathcal{Y}, \Sigma_{\mathcal{Y}})$, function classes $\mathcal{F} \subseteq \mathcal{M}_{\mathcal{X}, \mathcal{Y}}$, non-decreasing sequences $\{d_t : t \in \mathbb{N}\}$, any $T \in \mathbb{N}$ and $\epsilon > 0$:*

$$\sum_{k=1}^{M} \sum_{i=1}^{\tau} \mathbb{1}\{r_{\mathcal{F}_k}(x_{t_k+i}) > \epsilon\} < \left(\frac{d_T}{\tau \epsilon^2} + 1\right) 2\tau |\mathcal{X}|$$

*Proof.* By construction of $\mathcal{F}_t$ and noting that $d_t$ is non-decreasing in $t$, we can say that $r_{\mathcal{F}_k}(x_t) \leq \sqrt{d_T/n_{t_k}(x_t)}$ for all $t = 1, .., T$ so that

$$\sum_{k=1}^{M} \sum_{i=1}^{\tau} \mathbb{1}\{r_{\mathcal{F}_k}(x_{t+k+1}) > \epsilon\} \leq \sum_{k=1}^{M} \sum_{i=1}^{\tau} \mathbb{1}\left\{\sqrt{d_T/n_{t_k}(x_{t_k+i})} > \epsilon\right\}.$$

Now let $g(\epsilon) = \sqrt{d_T \epsilon^2/(d_T - \tau \epsilon^2)}$ be the $\epsilon$-inverse of $h^{(\tau)}(d_T, \epsilon)$ such that $g(h^{(\tau)}(d_T, \epsilon)) = \epsilon$. Applying Corollary 3 to our expression $n$ times repeatedly we can say:

$$\sum_{k=1}^{M} \sum_{i=1}^{\tau} \mathbb{1}\left\{\sqrt{d_T/n_{t_k}(x_{t_k+i})} > \epsilon\right\} \leq \sum_{k=1}^{M} \sum_{i=1}^{\tau} \mathbb{1}\left\{\sqrt{d_T/n_{t_k}(x_{t_k+i})} > g^{(n)}(\epsilon)\right\} + 2n\tau|\mathcal{X}|.$$

Where $g^{(n)}(\epsilon)$ denotes the composition of $g(\cdot)$ $n$-times acting on $\epsilon$. If we take $n$ to be the lowest integer such that $g^{(n)}(\epsilon) > \sqrt{d_T/\tau}$ then, $\sum_{k=1}^{M}\sum_{i=1}^{\tau} \mathbb{1}\left\{\sqrt{d_T/n_{t_k}(x_{t_k+i})} > g^{(n)}(\epsilon)\right\} \leq 2\tau|\mathcal{X}|$ so that the whole expression is bounded by $(n+1)\,2\tau|\mathcal{X}|$. Note that for all $N \in \mathbb{R}_+$, $g(\sqrt{d_T/N}) = \sqrt{d_T/(N-\tau)}$, if we write $\epsilon = \sqrt{d_T/N_1}$ then $n \leq N_1/\tau = \frac{d_T}{\tau\epsilon^2}$, which completes the proof.

$\square$

Using these results we are finally able to complete our proof of Theorem 3 We first note that, via the triangle inequality $\sum_{k=1}^{M}\sum_{i=1}^{\tau} w_{\mathcal{F}_k}(x_{t_k+i}) \leq 2\sum_{k=1}^{M}\sum_{i=1}^{\tau} r_{\mathcal{F}_k}(x_{t_k+i})$. We streamline our notation by letting $r_{k,i} = r_{\mathcal{F}_k}(x_{t_k+i})$. Reordering the sequence $(r_{1,1},..,r_{M,\tau}) \rightarrow (r_{i_1},..,r_{i_T})$ such $r_{i_1} \geq .. \geq r_{i_T}$ we have that:

$$\sum_{k=1}^{M}\sum_{i=1}^{\tau} r_{\mathcal{F}_k}(x_{t_k+i}) = \sum_{t=1}^{T} r_{i_t} \leq 1 + \sum_{i=1}^{T} r_{i_t}\mathbb{1}\{r_{i_t} \geq T^{-1}\}.$$

We can see that $r_{i_t} > \epsilon \geq T^{-1} \iff \sum_{i=1}^{T}\mathbb{1}\{r_{i_t} \geq \epsilon\} \geq t$. From Lemma 5 this means that $t \leq \left(\frac{d_T}{\tau\epsilon^2}+1\right)2\tau|\mathcal{X}|$, so that $\epsilon \leq \sqrt{\frac{2|\mathcal{X}|d_T}{t-2\tau|\mathcal{X}|}}$. This means that $r_{i_t} \leq \min\{C_{\mathcal{F}}, \sqrt{\frac{2|\mathcal{X}|d_T}{t-2\tau|\mathcal{X}|}}\}$. Therefore,

$$\begin{aligned}
\sum_{i=1}^{T} r_{i_t}\mathbb{1}\{r_{i_t} \geq T^{-1}\} &\leq 2\tau C_{\mathcal{F}}|\mathcal{X}| + \sum_{t=2\tau|\mathcal{X}|+1}^{T}\sqrt{\frac{2d_T|\mathcal{X}|}{t-\tau|\mathcal{X}|}} \\
&\leq 2\tau C_{\mathcal{F}}|\mathcal{X}| + \int_0^T \sqrt{\frac{2d_T|\mathcal{X}|}{t}}\,dt \\
&\leq 2\tau C_{\mathcal{F}}|\mathcal{X}| + 2\sqrt{2d_T|\mathcal{X}|T}
\end{aligned}$$

Which completes the proof of Theorem 3.

# B  Clean bounds for the symmetric problem

We now provide concrete clean upper bounds for Theorems 1 and 2 in the simple symmetric case $l+1 = m$, $C = \sigma = 1$, $|\mathcal{S}_i| = |\mathcal{X}_i| = K$ and $|Z_i^R| = |Z_i^P| = \zeta$ for all suitable $i$ and write $J = K^\zeta$. For a non-trivial problem setting we assume that $K \geq 2$, $m \geq 2$, $\tau \geq 2$.

From Section 7.3 we have that

$$\begin{aligned}
\mathbb{E}\left[\text{Regret}(T,\pi_\tau^{\text{PS}},M^*)\right] &\leq 4 + 2\sqrt{T} + m\left\{4(\tau J+1) + 4\sqrt{8\log(4mJT^2/\tau)JT}\right\} \\
&\quad + \mathbb{E}[\Psi]\left(1+\frac{4}{T-4}\right)m\left\{4(\tau J+1) + 4\sqrt{8K\log(4mJT^2/\tau)JT}\right\}
\end{aligned}$$

Through looking at the constant term we know that the bounds are trivially satisfied for all $T \leq 56$, from here we can certainly upper bound $4/(T-4) \leq 1/13$. From here we can say that:

$$\begin{aligned}
\mathbb{E}\left[\text{Regret}(T,\pi_\tau^{\text{PS}},M^*)\right] &\leq \left\{4 + 4m\left(1+\frac{14}{13}\mathbb{E}[\Psi]\right)(\tau J+1)\right\} \\
&\quad + \sqrt{T}\left\{2 + 4\sqrt{8J\log(4mJT^2/\tau)} + 4\sqrt{8JK\log(4mJT^2/\tau)}\frac{14}{13}\mathbb{E}[\Psi]\right\} \\
&\leq 5(1+\mathbb{E}[\Psi])\,m\tau J + \sqrt{T}\left\{12\sqrt{J\log(2mJT)} + 12\mathbb{E}[\Psi]\sqrt{JK\log(2mJT)}\right\} \\
&\leq 5(1+\mathbb{E}[\Psi])\,m\tau J + 12m\left(1+\mathbb{E}[\Psi]\sqrt{K}\right)\sqrt{JT\log(2mJT)} \\
&\leq \min(5m\tau^2 J, T) + 12m\tau\sqrt{JKT\log(2mJT)} \\
&\leq 15m\tau\sqrt{JKT\log(2mJT)}
\end{aligned}$$

Where in the last steps we have used that $\Psi \leq \tau$ and $\min(a,b) \leq \sqrt{ab}$. We now repeat a similar procedure of upper bounds for UCRL-Factored, immediately replicating $D$ by $\tau$ in our analysis to

say that with probability $\geq 1 - 3\delta$:

$$
\begin{aligned}
\text{Regret}(T, \pi_\tau^{\text{UC}}, M^*) &\leq \tau\sqrt{2T\log(2/\delta)} + 2\sqrt{T} + m\left\{4(\tau J + 1) + 4\sqrt{8\log(4mJT/\delta)JT}\right\} \\
&\quad + \tau m\left\{4(\tau J + 1) + 4\sqrt{8K\log(4mJT/\delta)JT}\right\} \\
&\leq (1+\tau)m4(\tau J + 1) + \\
&\quad \sqrt{T}\left\{\tau\sqrt{2\log(2/\delta)} + 2 + m4\sqrt{8\log(4mJT/\delta)J} + \tau m4\sqrt{8\log(4mJT/\delta)JK}\right\} \\
&\leq 5(1+\tau)m\tau J + 12m(1 + \tau\sqrt{K})\sqrt{JT\log(4mJT/\delta)} \\
&\leq 15m\tau\sqrt{JKT\log(4mJT/\delta)}
\end{aligned}
$$

Where in the last step we used a similar argument