[Reviews · NeurIPS 2014]

Submitted by Assigned_Reviewer_28

The paper proposes extensions to two existing algorithms, UCRL and PSRL, to take advantage of factored representations of MDPs. Regret bounds are derived for both algorithms, a high probability version for UCRL-factored, which seem to be the first such results for factored MDPs to the best of my knowledge. I believe that these results would be very influential in enabling more work in this area.

The paper is well written, though it is more comprehensible only when read with the supplementary material. There is possibly too much material being presented in a short paper. Another issue that I had with the presentation is that the results were presented before the algorithms themselves.

One thing that would add to the paper is some discussion on when each of the methods should be preferred in practice. Specifically some experimental results would have been useful.
Summary: First analysis of regret bounds for factored MDPs. A very comprehensive work that should be impactful.

Submitted by Assigned_Reviewer_42

This paper introduces two near-optimal RL algorithms for factored MDPs (FMDPs) with polynomial regret bounds. There's a significant literature on polynomial sample complexity in FMDPs, but this paper is the first, to the best of my knowledge and according to the authors claim, to show polynomial regret bounds.

This paper is well-written and technically sound, and it's an important result in this literature, even if mostly theoretical. One major drawback is the lack of experiments showing how, in practice, these two algorithms compare against each other and against polynomial sample complexity ones (like Factored RMax and others). Without these experiments, it's very hard to tell whether the relevance of this work is just theoretical or beyond.

The authors conclude that one drawback is the assumption that the algorithm has access to either a planning oracle or an appropriate approximate FMDP planner, but this is a problem common to all of this literature. The assumption that the algorithm has access to the factored structure of the MDP, or that it uses a structure learner (either the one cited or the simpler version with a tighter bound, http://machinelearning.org/archive/icml2009/papers/302.pdf), is also a common one. Again, it would be interesting if the authors tried this in a set of experiments.

Summary: This paper introduces two RL algorithms with provably polynomial regret bounds in Factored MDP. This is an important result, mostly theoretical, although the paper could be significantly improved by providing experiments showing how relevant the result is in practice.

Submitted by Assigned_Reviewer_44

Summary:

This paper introduces two algorithms for undiscounted, episodic RL in factored MDPs. The algorithms are modifications of the known RL algorithms UCRL2 and PSRL and assume that the underlying graph structure of the MDP is known. For both algorithms upper bounds on the regret are derived showing that the underlying structure can be exploited to give improved bounds compared to the original algorithms.

Evaluation:

This paper makes some progress with respect to exploiting structure in RL in MDPs. Algorithm and analysis are based on known techniques, so that the paper does not contribute much with respect to new methods. The paper is reasonably well-written, however it is not easy to read and appreciate the results, as there is a lot of notation, and the authors have tried to put a lot of material into the paper that maybe would rather deserve a journal publication.

Some (mostly minor) comments:

- The format seems not to be current NIPS format (but in the correct format the paper would rather be shorter).

- In the introduction, it is mentioned that there are sample-complexity bounds for factored MDPs in the literature. How do these compare to the regret bounds of the paper at hand?

- After the definition of regret, it is noted that the latter is random due to the randomness of M* (and the randomness of the algorithms and observations). It is not clear to me why M* is supposed to be random and not a fixed underlying MDP.

- In the definition of the factored MDPs I did not understand the role of the set X. Does this correspond to a set of state-action pairs?

- In Lemma 2, the notation P^*, \hat{P}_t, and n_t is not explained. Partly, this probably corresponds to some notation introduced before, but the lemma seems to be general.

Some typos:
- l.57: "acheive" should be "achieve"
- l.257: "optimistic over": something's missing here
- l.260: "modification to UCRL2 exploit": something's missing here
- l.332: "that may" should be "that we may"
- l.356: "we to bound" should be "we bound"
- l.424: "unadressed" should be "unaddressed"
Summary: Reasonable paper that makes some progress in exploiting structure in RL, yet without introducing any new methods.

Submitted by Assigned_Reviewer_45

New regret bounds for the problem of online RL in factored MDPs have been presented. The bounds improve on the-state-of-the-art in generic RL in terms of dependency on the size of MDP assuming that there exists some known graphical structure for the problem. For highly structured MDPs with local transitions this can lead to an exponential improvement in terms of dependency on the size of problem.

Quality and clarity

The paper is clearly written. I also quickly checked the technical parts ( proof sketches) which seems correct to me. As for the clarity of results I think the main trick to generalize the results of generic RL to the case of factored MDP have been clearly explained. In regard to the quality of results, I think the idea of exploiting the graphical structure to improve the regret bounds of RL seems a very promising idea to deal with large scale MDPs. However some of the assumptions such as the one that requires the prior knowledge of graphical structure seems to be a bit restrictive, as it is recognized by the authors, and may limit the applicability of this result. The paper would benefit from a more in-depth discussion of this issue. In particular a discussion on the possibility of combining the standard structure discovery algorithms with UCRL or PSRL would greatly add to the quality of the paper (also whether it is possible to prove similar regret bounds in that case). Finally the paper claims that the result is "near-optimal" however no corresponding lower bound to support this claim is provided.

Significance and novelty

The problem of exploration-exploitation trade off in factored MDPs has been extensively studied before in the RL literature . In that sense the idea of using the graphical structure to deal with large scale problems is not new. The main novelty is that here the bounds are expressed in terms of accumulated regret, whereas the focus of those prior works are mostly on PAC exploration bounds. This seems a rather incremental contribution since it is known that one can transform PAC complexity bounds to regret bounds and vice versa (See Jasch et al 10 and NIPS 11 workshop talk by Peter Auer). As for the novelty of the proof techniques, the paper mainly relies on the analysis of UCRL and PSRL and the main lemma to extend those results to the case of factored MDP is a standard algebraic trick.

Minor comments:

line 140: The connection between X and S and A should be defined earlier ( before definition 3).
line 298: where appropriate-> where it is appropriate
line 308-309: M^* and M_k are equal in law : this is only true when we use the true prior (prior from which M^* is sampled) in Bayesian inference. If we use a wrong prior for M_k then they are not any more equal.
Summary: The paper extends the regret bounds of UCRL and PSRL to the case of factored MDPs. The contribution w.r.t the-state-of-the-art is rather incremental. Also there is the issue of the practicality of the algorithms.
Author Feedback
Author rebuttal: Thank you very much for your feedback, we will try to deal with the main points in review.

1. Simulation results
**************************************
We considered adding simulation results for our algorithms, but did not include them for two reasons. First, approximate planning in factored MDPs is itself a non-trivial problem; any explanation of which might distract from the thrust of our paper. Second, we were already pressed for space in a short paper. However, it seems now this may not have been the correct choice and it might be a good idea to include some toy examples for the final version.

2. Comparison to existing methods
**************************************
In the paper we discuss the results of other analyses that lead to polynomial bounds at a relatively high level without too much detail. We do this deliberately because even to state the exact conditions and results of these analyses precisely might take up too much space and distract from our paper's message. At a high level we believe that regret bounds offer stronger guarantees and that our analyses are tighter although the problem settings are slightly different. We should maybe emphasize this point, or demonstrate with simulation.

3. Why is M* random?
**************************************
We present two algorithms with similar guarantees, but slightly different formulations. In particular, PSRL has bounds on the expected (Bayesian) regret where M* is randomly distributed according to the prior. We should take more care to distinguish the settings and results for PSRL and UCRL-Factored particularly to state what is random in each guarantee.

4. Other minor comments
**************************************
- Our formatting does seem to us to be the current NIPS style but we will double check for final submission.
- Lemma 2 should be expanded to reiterate what we mean by P* (the true distribution) and \hat{P}_t (the empirical distribution at t)